# A Novel Information Theoretic Objective to Disentangle Representations for Fair Textual Classification

## Abstract

One of the pursued objectives of deep learning is to provide tools that learn abstract representations of reality from the observation of multiple contextual situations. More precisely, one wishes to extract *disentangled representations* which are *(i)* low dimensional and *(ii)* whose components are independent and correspond to concepts capturing the essence of the objects under consideration (Locatello et al., 2019b). One step towards this ambitious project consists in learning disentangled representations *with respect to a predefined (sensitive) attribute, e.g.*, the gender or age of the writer. Perhaps one of the main application for such disentangled representations is fair classification. Existing methods extract the last layer of a neural network trained with a loss that is composed of a cross-entropy objective and a disentanglement regularizer. In this work, we adopt an information-theoretic view of this problem which motivates a novel family of regularizers that minimizes the mutual information between the latent representation and the sensitive attribute conditional to the target. The resulting set of losses, called `CLINIC`, is *parameter free* and thus, it is easier and faster to train. `CLINIC` losses are studied through extensive numerical experiments by training over 2k neural networks. We demonstrate that our methods offer a better disentanglement/accuracy trade-off than previous techniques, and generalize better than training with cross-entropy loss solely provided that the disentanglement task is not too constraining.

## 1 Introduction

There has been a recent surge towards disentangled representations techniques in deep learning Mathieu et al. (2019); Locatello et al. (2019a; 2020); Gabbay & Hoshen (2019). Learning disentangled representations from high dimensional data ultimately aims at separating a few explanatory factors (Bengio et al., 2013) that contain meaningful information on the objects of interest, regardless of specific variations or contexts. A disentangled representation has the major advantage of being less sensitive to accidental variations (*e.g.*, style) and thus generalizes well.

In this work, we focus on a specific disentanglement task which aims at learning a representation independent from a predefined attribute $S$. Such a representation will be called *disentangled with respect to $S$*. This task can be seen as a first step towards the ideal goal of learning a perfectly disentangled representation. Moreover, it is particularly well-suited for fairness applications, such as fair classification, which are nowadays increasingly sought after. When a learned representation $Z$ is disentangled from a sensitive attribute $S$ such as the age or the gender, any decision rule based on $Z$ is independent of $S$, making it fair in some sense. For instance, it satisfies the Demographic Parity criterion (see Proposition 1)).

Learning disentangled representations with respect to a sensitive attribute is challenging and previous works in the Natural Language Processing (NLP) community were based on two types of approach. The first one consists in training an adversary Elazar & Goldberg (2018); Coavoux et al. (2018). Despite encouraging results during training, Lample et al. (2018) show that a new adversary trained from scratch on the obtained representation is able to infer the predefined attribute, suggesting that the representation is in fact not disentangled. The second one consists in training a variational surrogate Cheng et al. (2020); Colombo et al. (2021); John et al. (2018) of the mutual information (MI) Cover & Thomas (2006) between the learned

representation and the variable from which one wishes to disentangle it. One of the major weaknesses of both approaches is the presence of an additional optimization loop designed to learn additional parameters during training (the parameters of the adversary for the first method, and the parameters required to approximate the MI for the second one), which is time-consuming and requires careful tuning (see Alg. 1).

**Our contributions.** We introduce a new method to learn disentangled representations, with a particular focus on fair representations in the context of `NLP`. Our contributions are two-fold:

**(1)** We provide **new perspectives on the disentanglement with respect to a predefined attribute problem** using information-theoretic concepts. Our analysis motivates the introduction of new losses tailored for classification, called `CLINIC` (**C**onditional mutua**L** **I**nformatio**N** m**I**nimization for fair **C**lassif**I**c**A**tio**N**). It is faster than previous approaches as it does not to require to learn additional parameters. One of the main novelty of `CLINIC` is to minimize the MI between the latent representation and the sensitive attribute *conditional to the target* which leads to high disentanglement capability while maintaining high-predictive power.

**(2)** We conduct **extensive numerical experiments** which illustrate and validate our methodology. More precisely, we train over 2K neural models on four different datasets and conduct various ablation studies using both Recurrent Neural Networks (`RNN`) and pre-trained transformers (`PT`) models. Our results show that the `CLINIC`'s objective is better suited than existing methods, it is faster to train and requires less tuning as it does not have learnable parameters. Interestingly, in some scenarios, it can increase both disentanglement and classification accuracies and thus, overcoming the classical disentanglement-accuracy trade-off.

From a practical perspective, we would like to add that our method is well-suited for fairness applications and is compliant with the *fairness through unawareness principle* that obliges one to fix a single model which is later applied across all groups of interest Gajane & Pechenizkiy (2017); Lipton et al. (2018). Indeed, our method only requires access to the sensitive attribute during its learning phase, but the resulting learned prediction function does not take the sensitive variable as an input.

## 2 Related Works

In order to describe existing works, we begin by introducing some useful notations. From an input textual data represented by a random variable (r.v.) $X \in \mathcal{X}$, the goal is to learn a parameter $\theta \in \Theta$ of an encoder $f_\theta : \mathcal{X} \to \mathcal{Z} \subset \mathbb{R}^d$ so as to transform $X$ into a latent vector $Z = f_\theta(X)$ of dimension $d$ that summarizes the useful information of $X$. Additionally, we require the learned embedding $Z$ to be *guarded* (following the terminology of Elazar & Goldberg (2018)) from an input sensitive attribute $S \in \mathcal{S}$ associated to $X$, in the sense that no classifier can predict $S$ from $Z$ better than a random guess. The final decision is done through predictor $g_\phi$ that makes a prediction $\hat{Y} = g_\phi(Z) \in \mathcal{Y}$, where $\phi \in \Phi$ refers to the learned parameters. We will consider classification problems where $\mathcal{Y}$ is a discrete finite set.

### 2.1 Disentangled Representations

The main idea behind most of the previous works focusing on the learning of disentangled representations consists in adding a disentanglement regularizer to a learning task objective. The resulting mainstream loss takes the form:

$$\mathcal{L}(\theta, \phi) = \underbrace{\mathcal{L}_{task}}_{\text{target task}} + \lambda \cdot \underbrace{\mathcal{R}\left(f_\theta(X), S; \phi\right)}_{\text{disentanglement}}, \tag{1}$$

where $\phi$ denotes the trainable parameters of the regularizer. Let us describe the two main types of regularizers used in the literature, both having the flaw to require a nested loop, which adds an extra complexity to the training procedure (see Alg. 1).

**Adversarial losses.** They rely on fooling a classifier (the adversary) trained to recover the sensitive attribute $S$ from $Z$. As a result, the corresponding disentanglement regularizer is trained by relying on the cross-entropy (`CE`) loss between the predicted sensitive attribute and the ground truth label $S$. Despite encouraging results, adversarial methods are known to be unstable both in terms of training dynamics Sridhar et al. (2021); Zhang et al. (2019) and initial conditions Wong et al. (2020).

**Losses based on Mutual Information (MI).** These losses, which also rely on learned parameters, aim at

minimizing the MI $I(Z; S)$ between $Z$ and $S$, which is defined by

$$I(Z; S) = \mathbb{E}_{ZS} \left[ \log \frac{p_{ZS}(Z, S)}{p_Z(Z)p_S(S)} \right], \tag{2}$$

where the joint probability density function (pdf) of the tuple $(Z, S)$ is denoted $p_{ZS}$ and the respective marginal pdfs are denoted $p_Z$ and $p_S$. Recent MI estimators include `MINE` (Belghazi et al., 2018), `NWJ` (Nguyen et al., 2010), `CLUB` (Cheng et al., 2020), `DOE` (McAllester & Stratos, 2020), $I_\alpha$ (Colombo et al., 2021), `SMILE` Song & Ermon (2019).

## 2.2 Parameter Free Estimation of MI

The `CLINIC`'s objective can be seen as part of the second type of losses although **it does not involve additional learnable parameters**. MI estimation can be done using contrastive learning surrogates Chopra et al. (2005) which offer satisfactory approximations with theoretical guarantees (we refer the reader to Oord et al. (2018) for further details). Contrastive learning is connected to triplet loss Schroff et al. (2015) and has been used to tackle the different problems including self-supervised or unsupervised representation learning (*e.g.* audio Qian et al. (2021), image Yamaguchi et al. (2019), text Reimers & Gurevych (2019); Logeswaran & Lee (2018)). It consists in bringing closer pairs of similar inputs, called *positive pairs* and further dissimilar ones, called *negative pairs*. The positive pairs can be obtained by data augmentation techniques Chen et al. (2020) or using various heuristic (*e.g* similar sentences belong to the same document Giorgi et al. (2020), backtranslation Fang et al. (2020) or more complex techniques Qu et al. (2020); Gillick et al. (2019); Shen et al. (2020)). For a deeper dive in mining techniques used in `NLP`, we refer the reader to Rethmeier & Augenstein (2021).

One of the novelty of `CLINIC` is to provide a novel information theoretic objective tailored for fair classification. It incorporates both the sensitive and target labels in the disentanglement regularizer.

## 2.3 Fair Classification and Disentanglement

The increasing use of machine learning systems in everyday applications has raised many concerns about the fairness of the deployed algorithms. Works addressing fair classification can be grouped into three main categories, depending on the step at which the practitioner performs a *fairness intervention* in the learning process: (i) pre-processing Brunet et al. (2019); Kamiran & Calders (2012), (ii) in-processing Colombo et al. (2021); Barrett et al. (2019) and (iii) post-processing d'Alessandro et al. (2017) techniques (we refer the reader to Caton & Haas (2020) for exhaustive review). When the attribute for which we want to disentangle the representation is a sensitive attribute (*e.g.* gender, age, race), our method can be considered as an in-process fairness technique. This is because a perfectly disentangled representation achieves a widely used definition of fairness namely *Demographic Parity*, as stated by the following proposition, whose proof is deferred to the Ssec. A.1.

**Proposition 1.** *Suppose $Z$ is independent of $S$ and denote by $\hat{Y} = g_\phi(Z)$. Then the decision rule satisfies the Demographic Parity criterion, which requires that $\hat{Y}$ is independent of $S$, i.e. $\mathbb{P}(\hat{Y} = y \mid S = 0) = \mathbb{P}(\hat{Y} = y' \mid S = 1)$ for all $y, y' \in \mathcal{Y}$.*

# 3 Model and Training Objective

In this section, we introduce the new set of losses called `CLINIC` that is designed to learn disentangled representations. We begin with information theory considerations which allow us to derive a training objective, and then discuss the relation to existing losses relying on MI.

## 3.1 Analysis & Motivations

### 3.1.1 Problem Analysis.

When learning disentangled representations, the goal is to obtain a representation $Z$ that contains no information about a sensitive attribute $S$ but preserves the maximum amount of information between $Z$ and

the target label $Y$. We use Veyne diagrams in Fig. 1 to illustrate the situation. Notice that, for a given task, the MI between $Y$ and $S$ is fixed (*i.e* corresponding to $\mathscr{C}_Y \cap \mathscr{C}_S$). Therefore, any representation $Z$ that maximizes the MI with $Y$ (*i.e* corresponding to $\mathscr{C}_Y \cap \mathscr{C}_Z$) cannot hope to have a mutual information with $S$ lower than $I(Y; S)$. Informally, recalling that $Z = f_\theta(X)$, we would like to solve:

$$\max_{\theta \in \Theta} \; I(Z; Y) - \lambda \cdot I(Z; S), \tag{3}$$

where $\lambda > 0$ controls the magnitude of the penalization. Existing works (Elazar & Goldberg, 2018; Barrett et al., 2019; Coavoux et al., 2018) rely on the `CE` loss to maximize the first term, i.e., within the area of $\mathscr{C}_Z \cap \mathscr{C}_Y$ in Fig. 1, and either on adversarial or contrastive methods for minimizing the second term, i.e., the area of $\mathscr{C}_Z \cap \mathscr{C}_S$ in Fig. 1). The ideal objective is to maximize the area of $(\mathscr{C}_Z \cap \mathscr{C}_Y) \setminus \mathscr{C}_S$. We refer to Colombo et al. (2021) for connections between adversarial learning and MI and to Oord et al. (2018) for connections between contrastive learning and MI.

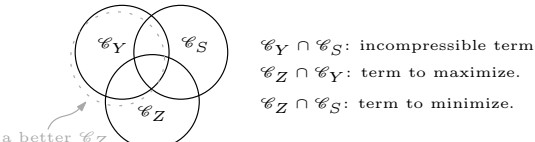

Fig. 1: Veyne diagrams visualization.

### 3.1.2 Limitations of Previous Methods

Since $I(Z; S) = I(Z; S|Y) + I(Z; S; Y)$, when minimizing $I(Z; S)$, previous work minimize actually the two terms $I(Z; S|Y)$ and $I(Z; S; Y)$. *This could be problematic since $I(Z; S; Y)$ tends to decrease the MI between $Z$ and $Y$, a phenomenon we would like to avoid to keep high performance on our target task.* Our method will bypass this issue by minimizing the conditional mutual information $I(Z; S|Y)$ solely (this amounts to minimize the area of $(\mathscr{C}_Z \cap \mathscr{C}_S) \setminus (\mathscr{C}_Z \cap \mathscr{C}_S \cap \mathscr{C}_Y)$.

### 3.2 `CLINIC`

Motivated by the previous analysis `CLINIC` aims at maximizing the new following ideal objective:

$$\max_{\theta \in \Theta} \; I(Z; Y) - \lambda \cdot I(Z; S|Y). \tag{4}$$

Minimizing $I(Z; S)$ in Eq. 4 instead of $I(Z; S)$ in Eq. 3 alleviates previously identified flaws.

### 3.2.1 Estimation of $I(Z; S|Y)$

The estimation of MI related quantities is known to be difficult Pichler et al. (2020); Paninski (2003). As a result, to estimate $I(Z; S|Y)$, we develop a tailored made constrastive learning objectives to obtain a parameter free estimator. Let us describe the general form for the loss we adopt on a given input $\{x_i, y_i, s_i\}_{1 \leq i \leq B}$ of size $B$. Recall that $z_i = f_\theta(x_i)$ is the output of the encoder for input $x_i$. For each $1 \leq i \leq B$, in fair classification task we have access to two subsets $\mathscr{P}(i), \mathscr{N}(i) \subset \{1, \ldots, B\} \setminus \{i\}$ corresponding respectively to *positive* and *negative* indices of examples. More precisely, $\mathscr{P}(i)$ (resp. $\mathscr{N}(i)$) corresponds to the set of indices $j \neq i$ such that $z_j$ is similar (resp. dissimilar) to $z_i$. Then, `CLINIC` consists in minimizing a loss of the form Eq. 1 with $\mathcal{L}_{task}$ given by the `CE` between the predictions $g_\phi(z_i)$ and the groundtruth labels $y_i$, and with $\mathcal{R}$ given by:

$$\mathcal{R} = \mathcal{R}(Z, \mathscr{P}, \mathscr{N}, B, \tau_p, \tau_n) = -\sum_{i=1}^{B} \mathrm{C}_i, \tag{5}$$

where the contribution $\mathrm{C}_i$ of sample $i$ is

$$\mathrm{C}_i = \frac{1}{|\mathscr{P}(i)|} \sum_{j_p \in \mathscr{P}(i)} \log \frac{e^{z_i \cdot z_{j_p} / \tau_p}}{\displaystyle\sum_{j_n \in \mathscr{N}(i)} e^{z_i \cdot z_{j_n} / \tau_n}}. \tag{6}$$

As emphasized in Eq. 5, the term $\mathcal{R}$ depends on several hyperparameters: the choice of positive and negative examples $(\mathcal{P}(i), \mathcal{N}(i))$, the associated temperatures $\tau_p, \tau_n > 0$ and the batch size $B$. Compared to Eq. 1, the proposed regularizer does not require any additional trainable parameters $\phi$.

### 3.2.2 Hyperparameters Choice

**Sampling strategy for $\mathscr{P}$ and $\mathscr{N}$.** The choice of positive and negative samples is instrumental for contrastive learning Wu et al. (2021); Karpukhin et al. (2020); Chen et al. (2020); Zhang & Stratos (2021); Robinson et al. (2020). In the context of fair classification, the input data take the form $(x_i, s_i, y_i)$ and we consider two natural strategies to define the subsets $\mathcal{P}$ and $\mathcal{N}$. For any given $y$ (resp. $s$), we denote by $\overline{y}$ (resp. $\overline{s}$) a uniformly sampled label in $\mathcal{Y} \setminus \{y\}$ (resp. in $\mathcal{S} \setminus \{s\}$).

**Remark 1.** *It is usual in fairness applications to consider that the sensitive attribute is binary. In that case $\bar{S}$ is deterministic.*

The first strategy $(\mathcal{S}_1)$ is to take

$$\mathcal{P}_{\mathcal{S}_1}(i) = \{1 \leq j_p \leq B, \;\; \text{s.t.} \;\; y_{j_p} = y_i, s_{j_p} = \bar{s}_i\},$$
$$\mathcal{N}_{\mathcal{S}_1}(i) = \{1 \leq j_n \leq B, \;\; \text{s.t.} \;\; y_{j_n} = \bar{y}_i\}.$$

The second strategy $(\mathcal{S}_2)$ is to set

$$\mathcal{P}_{\mathcal{S}_2}(i) = \{1 \leq j_p \leq B, \;\; \text{s.t} \;\; y_{j_p} = y_i, s_{j_p} = \bar{s}_i\},$$
$$\mathcal{N}_{\mathcal{S}_2}(i) = \{1 \leq j_n \leq B, \;\; \text{s.t} \;\; y_{j_n} = \bar{y}_i, s_{j_n} = s_i\}.$$

**Influence of the temperature.** As discussed in Wang & Liu (2021); Wang & Isola (2020) in the case where $\tau_p = \tau_n$, a good choice of temperature parameter is crucial for contrastive learning. Our method offers additional versatility by allowing to fine tune *two* temperature parameters, $\tau_p$ and $\tau_n$, respectively corresponding to the impact one wishes to put on positive and negative examples. For instance, a choice of $\tau_n \ll 1$ tends to focus on hard negative pairs while $\tau_n \gg 1$ makes the penalty uniform among the negatives. We investigate this effect in Ssec. 6.2.

**Influence of the batch size.** Previous works on contrastive losses Henaff (2020); Oord et al. (2018); Bachman et al. (2019); Mitrovic et al. (2020) argue for using large batch sizes to achieve good performances. In practice, hardware limits the maximum number of samples that can be stored in memory. Although several works He et al. (2020); Gao et al. (2021), have been conducted to go beyond the memory usage limitation, every experiment we conducted was performed on a single GPU. Nonetheless, we provide an ablation study with respect to admissible batch sizes in Sssec. B.3.1.

### 3.3 Theoretical Guarantees

There exists a theoretical bound between the contrastive loss of Eq. 5 and the mutual information between two probability laws in the latent space, defined according to the sampling strategy for $\mathscr{P}$ and $\mathscr{N}$. CLINIC's training objective offers theoretical guarantees when approximating $I(Z; S|Y)$ in Eq. 4. Formally, strategy $\mathcal{S}_1$ and $\mathcal{S}_2$ aim at minimizing the distance between:

$$\underbrace{P_Z(\cdot \mid Y = 0, S = 0)}_{\mathscr{L}_{0,0}} \text{ and } \underbrace{P_Z(\cdot \mid Y = 0, S = 1)}_{\mathscr{L}_{0,1}},$$

and between

$$\underbrace{P_Z(\cdot \mid Y = 1, S = 0)}_{\mathscr{L}_{1,0}} \text{ and } \underbrace{P_Z(\cdot \mid Y = 1, S = 1)}_{\mathscr{L}_{1,1}}.$$

We prove the following result in A.2.

**Theorem 1.** *For $\epsilon \in \{0,1\}$, denote by $p_\epsilon = |\{1 \leq i \leq B, \ y_i = \epsilon\}|/B$. Then, it holds that*

$$\frac{1}{p_0}I(\mathscr{L}_{0,0},\mathscr{L}_{0,1}) \quad + \quad \frac{1}{p_1}I(\mathscr{L}_{1,0},\mathscr{L}_{1,1}) \qquad \geq \qquad \frac{\log(p_0 B)}{p_0} \quad + \quad \frac{\log(p_1 B)}{p_1} \quad - \quad (\mathcal{R}/B). \quad (7)$$

**Remark 2.** *Th. 1 offers theoretical guarantees that our adaptation of contrastive learning in Eq. 5 is a good approximation of $I(Z;S|Y)$ in Eq. 4.*

**Remark 3.** *To simplify the exposition we restricted to binary $Y$ and $S$. The general case would involved quantities $\mathscr{L}_{y,s}$ with $y \in \mathcal{Y}$ and $S \in \mathcal{S}$.*

## 4 Experimental Setting

In this section, we describe the experimental setting which includes the dataset, the metrics and the different baselines we will consider. Due to space limitations, details on hyperparameters and neural network architectures are gathered in Ap. C. To ensure fair comparisons we re-implement all the models in a unified framework.

### 4.1 Datasets

We use the `DIAL` dataset Blodgett et al. (2016) to ensure backward comparison with previous works Colombo et al. (2021); Xie et al. (2017); Barrett et al. (2019). We additionally report results on `TrustPilot` (`TRUST`) Hovy et al. (2015) that has also been used in Coavoux et al. (2018). Tweets from the `DIAL` corpus have been automatically gathered and labels for both polarity (*is the expressed sentiment positive or negative?*) and mention (*is the tweet conversational?*) are available. Sensitive attribute related to the race (*is the author non-Hispanic black or non-Hispanic white?*) has been inferred from both the author geo-location and the used vocabulary. For `TRUST` the main task consists in predicting a sentiment on a scale of five. The dataset is filtered and examples containing both the author birth date and gender are kept and splits follow Coavoux et al. (2018). These variables are used as sensitive information. To obtain binary sensitive attributes, we follow Hovy & Søgaard (2015) where age is binned into two categories (*i.e.* age under 35 and age over 45). **A word on the sensitive attribute inference.** Notice that these two datasets are balanced with respect to the chosen sensitive attributes ($S$), which implies that a random guess has 50% accuracy.

### 4.2 Metrics

Previous works on learning disentangled representation rely on two metrics to assess performance.
**Measuring disentanglement** by reporting the accuracy of an adversary trained from scratch to predict the sensitive labels from the latent representation. Since both datasets are balanced, *a perfectly disentangled representation corresponds to an accuracy of the adversary of 50%.*
**Success on the main classification task** which is measured with accuracy (higher is better). As we are interested in controlling the desired degree of disentanglement (Colombo et al., 2021), we report the trade-off between these two metrics for different models when varying the $\lambda$ parameter, which controls the magnitude of the regularization (see Eq. 1). For our experiments, we choose $\lambda \in [0.001, 0.01, 0.1, 1, 10]$.

### 4.3 Baselines

**Losses.** To compare `CLINIC` with previous works, we compare against adversarial training (ADV) Elazar & Goldberg (2018); Coavoux et al. (2018); Barrett et al. (2019) and the recently introduced Mutual Information upper bound Colombo et al. (2021) ($I_\alpha$) which has been shown to offer more control over the degree of disentanglement than previous estimators. We compare `CLINIC` with the classical constrastive method that estimates $I(Z;S)$ (see Eq. 4). Beware that this baseline, be denoted as $\mathcal{S}_0$, does not incorporate information on $Y$.
**Encoders.** To provide an exhaustive comparison, we work both with `RNN`-encoder and `PT` (*e.g* `BERT` Devlin et al. (2018)) based architectures. Contrarily to previous works that use frozen `PT` Ravfogel et al. (2020), we fine-tune the encoder during training and evaluate our methods on various types of encoders (*e.g.* `DISTILBERT`

(`DIS.`) Sanh et al. (2019), `ALBERT` (`ALB.`) Lan et al. (2019), `SQUEEZEBERT` (`SQU.`) Iandola et al. (2020)). These models are selected based on efficiency.

## 5 Numerical Results

In this section, we gather experimental results on the fair classification task. Because of space constraints, additional results can be found in Ap. B.

### 5.1 Overall Results

We report in Tab. 1 the best model on each dataset for each of the considered methods. In Tab. 1, each row corresponds to a single $\lambda$ which controls the weight of the regularizer (see Eq. 1).

Tab. 1: Overall results on the fair classification task: the columns with $Y$ and $S$ stand for the main and the sensitive task accuracy respectively. ↓ means lower is better whereas ↑ means higher is better. The best model is bolded and second best is underlined. `CE` refers to a model trained based on `CE` solely (case $\lambda = 0$ in Eq. 1).

| Dat. | Loss | | RNN | | | BERT | |
|---|---|---|---|---|---|---|---|
| | | $\lambda$ | $Y(\uparrow)$ | $S(\downarrow)$ | $\lambda$ | $Y(\uparrow)$ | $S(\downarrow)$ |
| DIAL-S | CE | 0.0 | 62.7 | 73.2 | 0.0 | 76.2 | 76.7 |
| | $\mathcal{S}_0$ | 1.0 | 66.1 | 55.1 | 10 | 74.5 | 66.8 |
| | $\mathcal{S}_1$ | 1.0 | _79.1_ | _51.0_ | 1.0 | **73.5** | **58.3** |
| | $\mathcal{S}_2$ | 1.0 | **79.9** | **50.0** | 0.1 | _75.1_ | _69.4_ |
| | ADV | 1.0 | 58.4 | 70.2 | 0.1 | 74.8 | 72.7 |
| | $I_\alpha$ | 0.1 | 55.2 | 72.3 | 0.1 | 74.5 | 70.1 |
| DIAL-M | CE | 0.0 | 77.5 | 62.1 | 0.0 | 82.7 | 79.1 |
| | $\mathcal{S}_0$ | 10 | 76.7 | 54.9 | 10 | 76.6 | 66.6 |
| | $\mathcal{S}_1$ | 10 | **69.3** | **50.0** | 1.0 | _81.6_ | _52.0_ |
| | $\mathcal{S}_2$ | 10 | **69.3** | **50.0** | 1.0 | **75.0** | **50.0** |
| | ADV | 0.01 | 76.9 | 57.9 | 0.1 | 82.6 | 74.6 |
| | $I_\alpha$ | 0.1 | 75.5 | 55.7 | 10 | 74.9 | 55.0 |
| TRUST-A | CE | 0.0 | 72.9 | 53.0 | 0.0 | 74.9 | 53.8 |
| | $\mathcal{S}_0$ | 10 | 69.3 | 50.0 | 10 | 70.1 | 50.0 |
| | $\mathcal{S}_1$ | 10 | **75.1** | **50.0** | 10 | **75.1** | **50.0** |
| | $\mathcal{S}_2$ | 10 | _71.1_ | _50.0_ | 10 | **75.1** | **50.0** |
| | ADV | 10 | 65.5 | 50.0 | 10 | 70.1 | 50.0 |
| | $I_\alpha$ | 10 | 75.1 | 55.0 | 10 | 70.1 | 50.0 |
| TRUST-G | CE | 0.0 | 75.4 | 52.0 | 0.0 | 74.9 | 53.8 |
| | $\mathcal{S}_0$ | 10 | 73.1 | 50.0 | 10 | 73.3 | 50.0 |
| | $\mathcal{S}_1$ | 10 | **76.1** | **50.0** | 10 | **76.2** | **50.0** |
| | $\mathcal{S}_2$ | 10 | _74.8_ | _50.0_ | 10 | **76.2** | **50.0** |
| | ADV | 10 | 56.3 | 50.0 | 10 | 73.2 | 50.0 |
| | $I_\alpha$ | 10 | 76.2 | 50.0 | 10 | 73.2 | 50.3 |

**Global performance.** For each dataset, we report the performance of a model trained without disentanglement regularization (CE rows in Tab. 1). Results indicate this model relies on the sensitive attribute $S$ to perform the classification task. In contrast, all disentanglement techniques reduce the predictability of $S$ from the representation $Z$. Among these techniques, we observe that $I_\alpha$ improves upon ADV as already pointed out in Colombo et al. (2021). Our `CLINIC` based methods outperform both ADV and $I_\alpha$ baselines, suggesting that contrastive regularization is a promising line of search for future work in disentanglement.
**Comparing the strategies of `CLINIC`.** Among the three considered sampling strategies for positives and negatives, $\mathcal{S}_1$ and $\mathcal{S}_2$ are the best and always improve performance upon $\mathcal{S}_0$. This is because $\mathcal{S}_1$ and $\mathcal{S}_2$ incorporate knowledge on the target task to construct positive and negative samples, which is crucial to obtain good performance.
**Datasets difficulty.** From Tab. 1, we can observe that some sensitive/main label pairs are more difficult

to disentangle than others. In this regard, `TRUST` is clearly easier to disentangled than `DIAL`. Indeed, every models except for `CE` achieve perfect disentanglement. This suggests we are in the case $\mathscr{C}_Y \cap \mathscr{C}_S = \emptyset$ of Fig. 1, meaning that the sensitive attribute $S$ only contains few information on the target $Y$. Within `DIAL`, sentiment label is the hardest but `CLINIC` with strategy $\mathcal{S}_1$ achieves a good trade-off between accuracy and disentanglement.

**`RNN` vs `BERT` encoder.** Interestingly, the `BERT` encoder is always harder to disentangle than the `RNN` encoder. This observation can be seen as an additional evidence that `BERT` may exhibit gender, age and/or race biases, as already pointed out in Ahn & Oh (2021); Mozafari et al. (2020); de Vassimon Manela et al. (2021).

### 5.2   Controlling the Level of Disentanglement

A major challenge when disentangling representations is *to be able to control the desired level of disentanglement* Feutry et al. (2018). We report performance on the main task and on the disentanglement task for both `BERT` (see Fig. 2) and `RNN` (see Fig. 3) for differing $\lambda$. Notice that we measure the performance of the disentanglement task by reporting the accuracy metric of a classifier trained on the learned representation.

**Results on `DIAL`.** We report a different behavior when working either with `RNN` or `BERT`. From Fig. 3h we observe that `CLINIC` (especially $\mathcal{S}_1$ and $\mathcal{S}_2$) allows us to both learn perfectly disentangled representation as well as allow a fined grained control over the desirable degree of disentanglement when working with `RNN`-based encoder. On the other hand, previous methods (*e.g* ADV or $I_\alpha$) either fail to learn disentangled representations (see $I_\alpha$ on Fig. 2c) or do it while losing the ability to predict $Y$ (see ADV on Fig. 2c). As already pointed out in the analysis of Tab. 1, we observe again that it is both easier to learn disentangled representation and to control the desire degree of disentanglement with `RNN` compared to `BERT`.

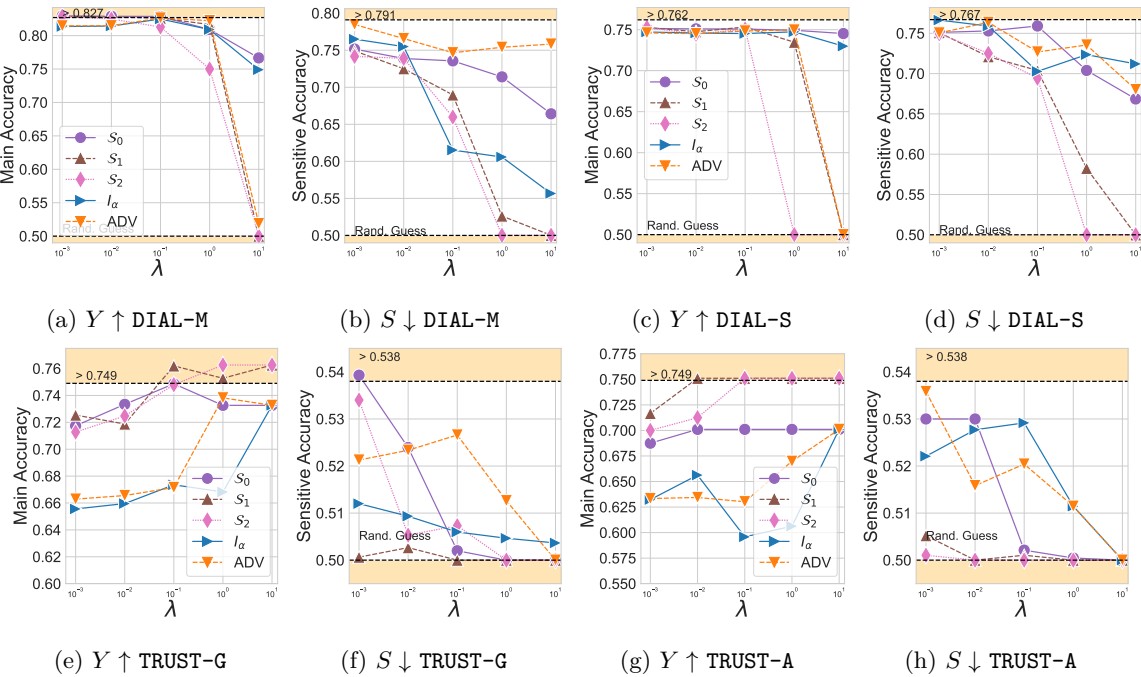

Fig. 2: Fair Classification results using `BERT`. Results are given with $B = 256$ and $\tau_p = \tau_n = 0.5$.

**Results on `TRUST`.** This dataset exhibits an interesting behaviour known as spurious correlations Yule (1926); Simon (1954); Pearl et al. (2000). This means that, without any disentanglement penalty, the encoder learns information about the sensitive features that hurts the classification performance on the test set. We also observe that learning disentangled representation with `CLINIC` (using $S_1$ or $S_2$) outperforms a model trained with `CE` loss solely. This suggests that `CLINIC` could *go beyond the standard disentanglement/accuracy trade-off.*

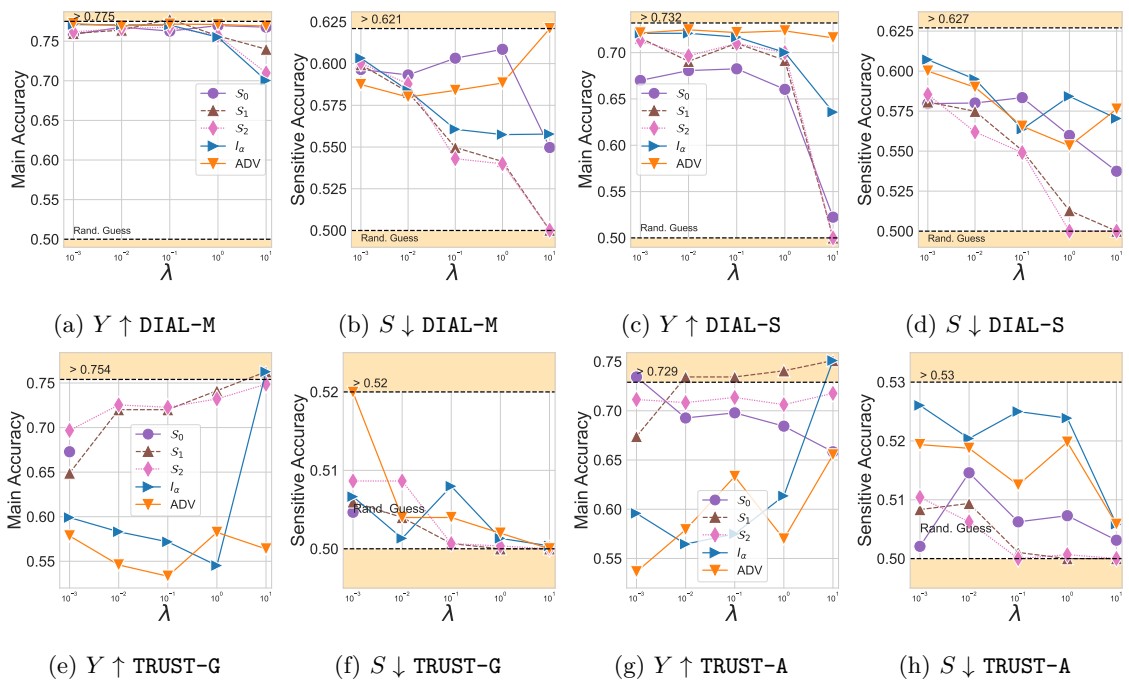

Fig. 3: Results on Fair Classification for a `RNN`.

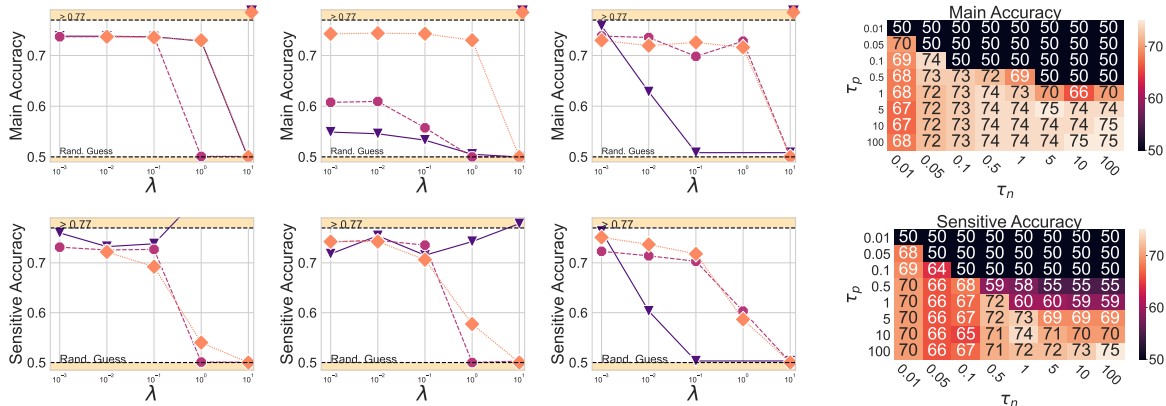

Fig. 4: Ablation study on `PT` on `DIAL-S`. Figures from left to right correspond to the performance of `ALB.`, `DIS.` and `SEQU.`.

Fig. 5: Ablation study on $(\tau_n, \tau_p)$ for `DIAL-S`.

### 5.3 Superiority of the `CLINIC`'s Objective

In this experiment we assess the relevance of using $I(Z; S|Y)$ (Eq. 4) instead of $I(Z; S)$ (Eq. 3). For all the considered $\lambda$, all datasets and all the considered checkpoints, we display in Fig. 6 the disentanglement/accuracy trade-off.

**Analysis.** Each point in Fig. 6 corresponds to a trained model (with a specific $\lambda$). The more a point is at the bottom right, the better it is for our purpose. Notice that the points stemming from our strategies $\mathcal{S}_1$ and $\mathcal{S}_2$ (orange/green) lie further down on the right than the point stemming from $\mathcal{S}_0$ (bleu). For instance, the use of $\mathcal{S}_1$ or $\mathcal{S}_2$ for the `RNN` provide many models exhibiting perfect sensitive accuracy while maintaining high main accuracy, which is not the case for $\mathcal{S}_0$. For `BERT`, models trained with $\mathcal{S}_0$ either have high sensitive and main accuracy or low sensitive and main accuracy. On the contrary, there are points stemming from $\mathcal{S}_1$ or $\mathcal{S}_2$ that lies on the bottom right of Fig. 6. *Overall, Fig. 6 validates the use of $I(Z; S|Y)$ instead of $I(Z; S)$.*

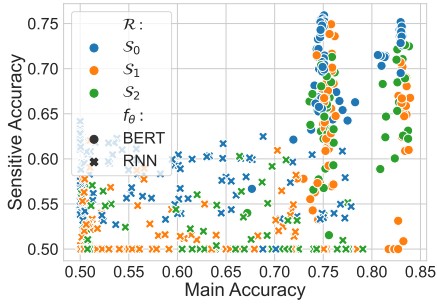

Fig. 6: Disentanglement/accuracy trade-off for various datasets, checkpoints and values of $\lambda$. Point with high sensitive accuracy (*e.g.* in the upper right corner) corresponds to low values of $\lambda$ (*e.g.* $\lambda \in \{10^{-3}, 10^{-2}, 10^{-1}\}$).

## 5.4 Speed up Gain

In contrast with `CLINIC`, both previous methods ADV and $I_\alpha$ rely on an additional network for the computation of the disentanglement regularizer in Eq. 1. These extra parameters need to be learned with the use of an additional loop during training: at each update of the encoder $f_\theta$, several updates of the network are performed to ensure that the regularizer computes the correct value. This loop is both times consuming and involves extra parameters (*e.g* new learning rates for the additional network) that must be tuned carefully. This makes ADV and $I_\alpha$ more difficult to implement on large-scale datasets. Tab. 2, illustrates both the parameter reduction and the speed up induced by `CLINIC`.

|      | Method | # params. | 1 epoch. |
|------|--------|-----------|----------|
| RNN  | ADV    | $2220$ $_{-0.6\%}$ | $551$ $_{-17\%}$ |
|      | $I_\alpha$ | $2234$ | $663$ |
|      | CLINIC | $2206$ $_{-1.3\%}$ | $500$ $_{-24\%}$ |
| BERT | ADV    | $109576$ $_{-0.01\%}$ | $2424$ $_{-10\%}$ |
|      | $I_\alpha$ | $109591$ | $2689$ |
|      | CLINIC | $109576$ $_{-0.03\%}$ | $2200$ $_{-19\%}$ |

Tab. 2: Runtime for 1 epoch (using `DIAL-S`, $B = 64$ and relying on a single NVIDIA-V100 with 32GB of memory). The model sizes are given in thousand. We compute the relative improvement with respect to the strongest baseline $I_\alpha$ from Colombo et al. (2021).

# 6 Ablation Study

In this section, we conduct an ablation study on `CLINIC` with the best sampling strategy ($\mathcal{S}_1$) to better understand the importance of its relative components. We focus on the effect of *(i)* the choice of `PT` models, *(ii)* the batch size and *(iii)* the temperature. This ablation study is conducted for both `DIAL-S` and `TRUST-A`, where we recall that the former is harder to disentangle than the latter. Results on `TRUST-A` can be found in Ap. B.

## 6.1 Changing the `PT` Model

**Setting.** As recently pointed out by Bommasani et al. (2021), `PT` plays a central role in `NLP`, thus the need to understand their effects is crucial. We test `CLINIC` with `PT` that are lighter and require less computation time to finetune than `BERT`.
**Analysis.** The results of `CLINIC` trained with `SQU.`, `DIS.` and `ALB.` are given in Fig. 4. Overall, we observe that `CLINIC` consistently achieves better results on all the considered models. Interestingly, for $\lambda > 0.1$, we observe that ADV degenerates: the main task accuracy is around 50% and the sensitive task accuracy either is 50% or reaches a high value. This phenomenon has been reported in Barrett et al. (2019); Colombo et al. (2021).

## 6.2 Effect of the Temperature

Recall that `CLINIC` uses two different temperatures (see Eq. 5) denoted by $\tau_p$ and $\tau_n$, corresponding to the magnitude one wishes to put on positive and negatives examples. In this experiment, we study their relative importance on the disentanglement/accuracy trade-off.

**Analysis.** Fig. 5 gathers the performance of `CLINIC` for different $(\tau_p, \tau_n)$. We observe that low values of $\tau_p$ (*i.e* focusing on easy positive) conduct to uninformative representation (*i.e* low accuracy for $Y$). As $\tau_p$ increases, the choice of $\tau_n$ becomes relevant. Previously introduced supervised contrastive losses Khosla et al. (2020) only use one temperature thus can only rely on diagonal score from Fig. 5. Since the chosen trade-off depends on the final application, we believe this ablation study validates the use of two temperatures.

## 7 Summary and Concluding Remarks

We introduced `CLINIC`, a set of novel losses tailored for fair classification. `CLINIC` both outperform existing disentanglement methods and can go beyond the traditional accuracy/disentanglement trade-off. Future works include (1) improving `CLINIC` to enable finer control over the disentanglement degree, and (2) developing a way to measure the accuracy/disentanglement trade-off which appears to differ for each dataset.

## 8 Limitations

This paper proposes a novel information-theoretic objective to learn disentangled representations. While the results held for English and studied pre-trained encoders, we observed different behavior depending on the disentanglement difficulty. Overall predicting for which attribute or which data we will be able to see a positive trade-off while disentangling remains an open question.

Additionally, similarly to previous work in the same line of research we also assumed to have access to (a binary) $S$ which might not be the case for various practical applications.

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

# A Proofs

## A.1 Proof of Proposition 1

Recall $\hat{Y} = g_\phi(Z)$ and assume that $Z$ is independent of the sensitive attribute $S$. We have the following:

$$\begin{aligned}
\mathbb{P}(\hat{Y} = y \mid S = 0) &= \mathbb{P}(g_\phi(Z) = y \mid S = 0) \\
&= \frac{\mathbb{P}(g_\phi(Z) = y, \, S = 0)}{\mathbb{P}(S = 0)} \\
&= \frac{\mathbb{P}(g_\phi(Z) = y)\mathbb{P}(S = 0)}{\mathbb{P}(S = 0)} \\
&= \mathbb{P}(\hat{Y} = y),
\end{aligned}$$

where we use the independence assumption in the second equality. The same computation is valid for $\mathbb{P}(\hat{Y} = y' \mid S = 1)$ which gives the *Demographic Parity*:

$$\mathbb{P}(\hat{Y} = y \mid S = 0) = \mathbb{P}(\hat{Y} = y' \mid S = 1).$$

## A.2 Proof of Theorem 1

Let us define:

$$\mathcal{B}_0 := \{1 \leq i \leq B : y_i = 0\}$$

and

$$\mathcal{B}_1 := \{1 \leq i \leq B : y_i = 1\}.$$

The regularization term defined in Eq. 5 can be rewritten as follows:

$$\mathcal{R} = -\left(\sum_{i \in \mathcal{B}_0} C_i + \sum_{i \in \mathcal{B}_1} C_i\right).$$

Then, denoting by $B_0$ (resp. $B_1$) the cardinal of $\mathcal{B}_0$ (resp. $\mathcal{B}_1$) and following Oord et al. (2018), it holds that:

$$-\frac{1}{B_0} \sum_{i \in \mathcal{B}_0} C_i \geq \log(B_0) - I(\mathcal{L}_{0,0}, \mathcal{L}_{0,1})$$

and

$$-\frac{1}{B_1} \sum_{i \in \mathcal{B}_1} C_i \geq \log(B_1) - I(\mathcal{L}_{1,0}, \mathcal{L}_{1,1})$$

As a result,

$$- \quad (\mathcal{R}/B) \quad \geq \quad \frac{\log(B_0)}{p_0} \quad + \quad \frac{\log(B_1)}{p_1} \frac{I(\mathcal{L}_{0,0}, \mathcal{L}_{0,1})}{p_0} \quad - \quad \frac{I(\mathcal{L}_{1,0}, \mathcal{L}_{1,1})}{p_1},$$

from which we deduce the desired result.

## A.3 Additional Motivation

We provide in Fig. 7 example of extreme situations when learning to disentangle representations. This figure can be analyzed using the Venne diagram of Fig. 1.

## A.4 Algorithms used for the baseline models.

When training an adversary or a MI with learnable parameters the procedure requires extra learnable parameters that need to be tuned using a Nested Loop.

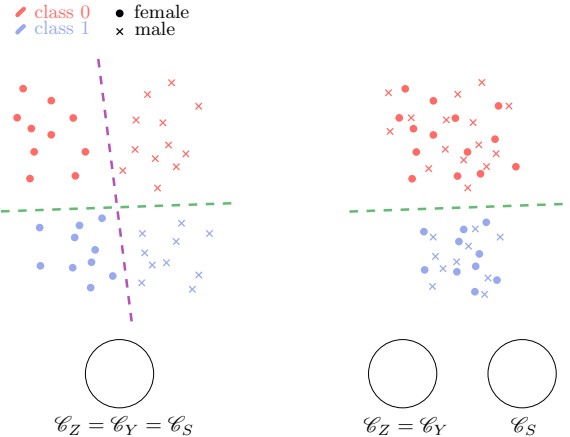

Fig. 7: Two extremes situations. In both cases, a classifier can predict the main task with high accuracy. The right representation is fairer than the left one as the gender is harder to extract.

---

**Algorithm 1** Procedure to learn the baselines

---

1: INPUT Labeled dataset $\mathcal{D} = \{(x_j, s_j, y_j), \forall j \in [1, |\mathcal{D}| - K]\}$, independant labeled dataset $\mathcal{D}' = \{(x_j, s_j, y_j), \forall j \in [|\mathcal{D}| - K, |\mathcal{D}|]\}$, $\theta$ parameters of the encoder network, $\phi$ parameters of the main classifier, $\psi$ parameters of the regularizer.
2: INITIALIZE parameters $\theta, \phi, \psi$
3: OPTIMIZATION
4: Freeze $\theta$
5: **while** $(\theta, \phi, \psi)$ not converged **do**
6:     **for** $i \in [1, Unroll]$ **do**                                                  ▷ Nested loop
7:         Sample a batch $\mathcal{B}'$ from $\mathcal{D}'$
8:         Update $\phi, \psi$ using (Eq. 1).
9:     **end for**
10:     Unfreeze $\theta$ and freeze $\phi, \psi$
11:     Sample a batch $\mathcal{B}$ from $\mathcal{D}$
12:     Update $\theta$ with $\mathcal{B}$ (Eq. 1).
13:     Unfreeze $\phi, \psi$
14: **end while**
15: OUTPUT Encoder and classifier weights $\theta, \phi$

---

## B  Additional Experimental Results

In this section, we report additional experiments. Specifically, we gather:

- All the unreported experiments in the core paper for fair classification task on both RNN and BERT for the four considered sensitive/main attribute pairs (see Ssec. B.2 for TRUST and Ssec. B.1 for DIAL).

- The ablation study on the PT choice for TRUST-G (see Sssec. B.3.2).

- The ablation study on the batch size for TRUST-G (see Sssec. B.3.1).

- The ablation study on the impact of the temperature for TRUST-G (see Sssec. B.3.3).

### B.1  Controlling experiment on DIAL

Important aspects when learning to disentangle representations are (*i*) achieving perfect independence between considered r.v while achieving high accuracy on the target task but also (*ii*) to control the desired level of

disentanglement Feutry et al. (2018). To ease visual comparison, we report the results on the two main attributes mention (`DIAL-M`) and sentiment (`DIAL-S`) while protecting the race.

**RNN-based encoder.** We report in Fig. 8 the results of the `RNN` encoder on `DIAL`. We observe that for both cases, `CLINIC` is able to achieve perfect disentanglement. The loss $I_\alpha$ plateau for $\lambda \geq 0.1$. Overall the two best strategies ($\mathcal{S}_1$ and $\mathcal{S}_2$) achieve stronger results than considered baselines.

**BERT-based encoder.** For `BERT` we gather the results in Fig. 9. Similarly to what is reported in Ssec. 5.2, we observe a steep transition and notice that all considered estimators provide little control over the desired degree of disentanglement. It is worth noting that for `DIAL-S`, $\mathcal{S}_2$ underperforms and is not able to achieve better than 69.4 on the sensitive task accuracy. This failure case has motivated our choice for selecting $\mathcal{S}_1$ while conducting the ablation studies.

**Takeaways.** Overall, we observe that `CLINIC` outperforms considered baselines when combined with both `RNN` and `BERT` encoders on the `DIAL` dataset.

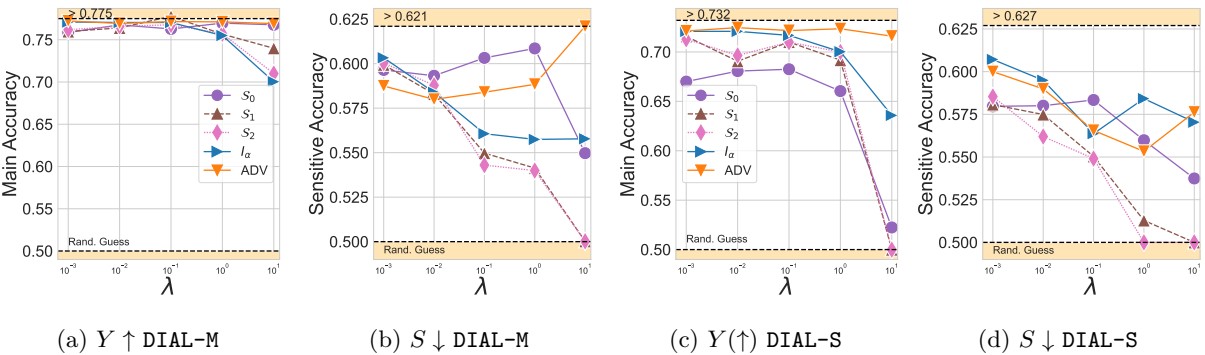

(a) $Y \uparrow$ `DIAL-M`     (b) $S \downarrow$ `DIAL-M`     (c) $Y(\uparrow)$ `DIAL-S`     (d) $S \downarrow$ `DIAL-S`

Fig. 8: Results on `DIAL` for the fair classification task relying on a `RNN` encoder.

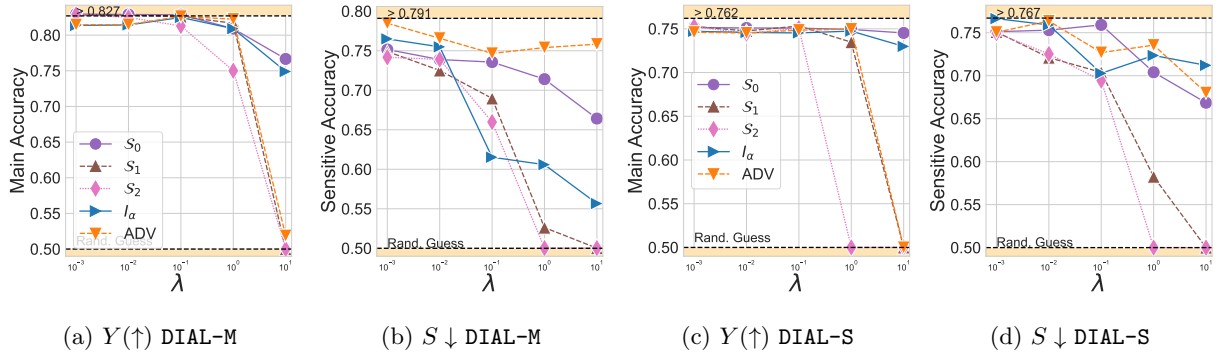

(a) $Y(\uparrow)$ `DIAL-M`     (b) $S \downarrow$ `DIAL-M`     (c) $Y(\uparrow)$ `DIAL-S`     (d) $S \downarrow$ `DIAL-S`

Fig. 9: Results on `DIAL` for the fair classification task using `BERT`.

### B.2 Controlling experiment on `TRUST`

Here we report the results on `TRUST` for both attribute age (`TRUST-A`) and gender (`TRUST-A`) while predicting the sentiment on a like scale of 5.

**RNN-based encoder.** From Fig. 10, we observe that both attributes (*i.e* gender and age) can be disentangled from the main attribute (*i.e* sentiment label). Interestingly, we observe a general trend: the more the representations are disentangled (*i.e* the lower the sensitive accuracy) the higher accuracy is obtained on the main task. On both `TRUST` datasets, we observe that both `CLINIC` and $I_\alpha$ are able to outperform the`CE` model (reported by the upper dash line). Overall, most of the models are able to achieve almost perfectly disentangled representations, *i.e* for all models with $\lambda = 10$ the sensitive accuracy falls below 51%.

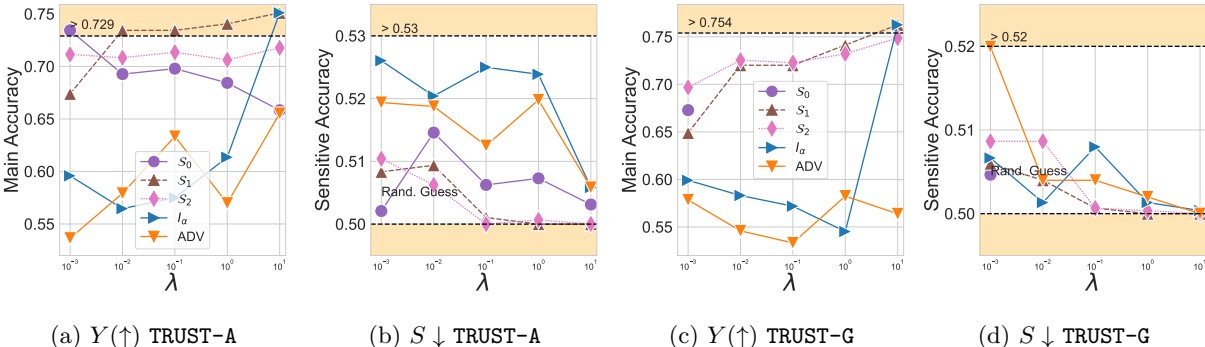

Fig. 10: Fair classification task on `TRUST` using a `RNN` encoder.

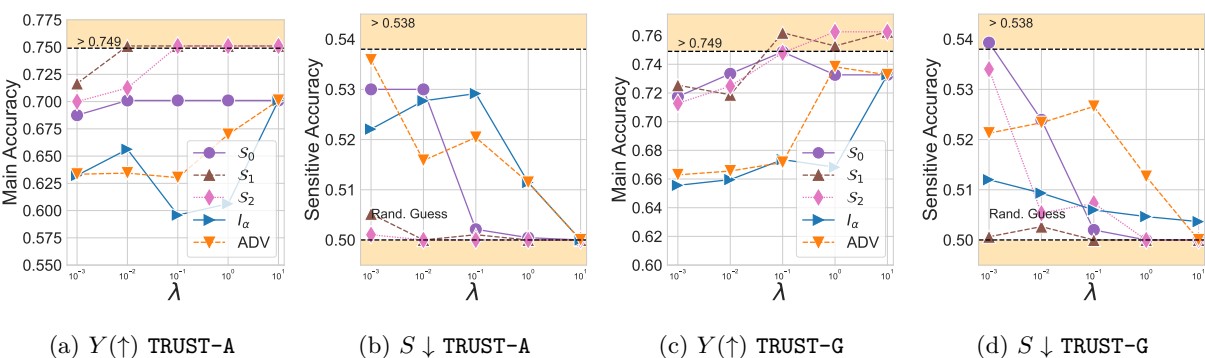

Fig. 11: Fair classification task results on `TRUST` with `BERT`.

**BERT-based encoder.** From Fig. 11, we observe similar trends to the previous ones: learning disentangled representation improves general performances on the main task. This phenomenon can be interpreted through the lens of spurious correlations Calude & Longo (2017). We observe that `CLINIC` with both $\mathcal{S}_1$ and $\mathcal{S}_0$ is able to remove almost all information from the sensitive labels even using small values of $\lambda$.

**Takeaways.** Overall on `TRUST`, it is easier to disentangle the representations from the sensitive attribute. We observe that `CLINIC` achieves strong results with both $\mathcal{S}_1$ and $\mathcal{S}_2$. It shows that incorporating the main label ($Y$) information in the sampling strategy for supervised contrastive learning loss is a key to achieve a good trade-off.

### B.3  Additional results on ablation studies

In this section, we gather the results of the ablation studies conducted on `CLINIC` to better understand the relative importance of the different elements. We specifically report results on `TRUST-A` on the batch size (Sssec. B.3.1), the choice of the `PT` model (Sssec. B.3.2) as well as the role of the temperature (Sssec. B.3.3).

### B.3.1  Effect of the batch size

**Setting.** As mentioned in Sec. 3, the batch size plays a key role when learning disentangled representations. In order to study its influence on `CLINIC`, we choose to work with `DIS.`, to fit large batch sizes on a single GPU.

**Analysis.** We experiment with $\lambda = 0.1$ and report the results on `DIAL` in Fig. 12. Interestingly, we observe a threshold phenomenon: for each model, a small value of the batch size leads to a poor performance on the main task. In contrast, working with large batch (*i.e* of size greater than 300) allows to learn representations that achieve low sensitive accuracy (around 57%) while maintaining a high main task accuracy (around 75%). On an easier task, changing the batch size does not impact the performances.

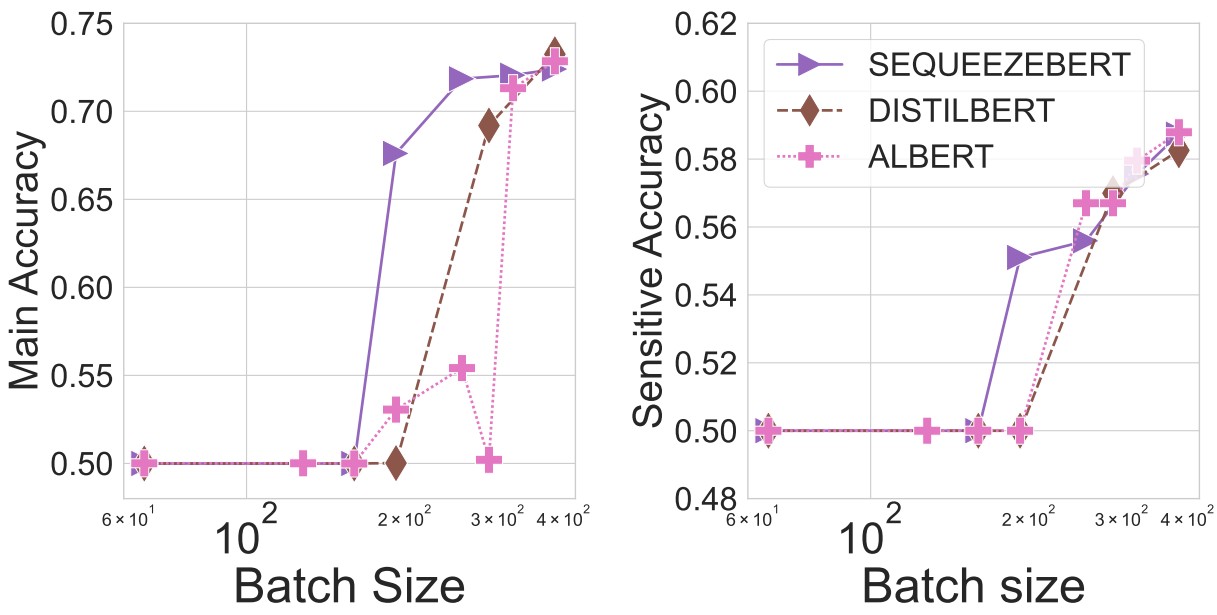

Fig. 12: Ablation on batch size on `DIAL-S` ($\lambda = 1$).

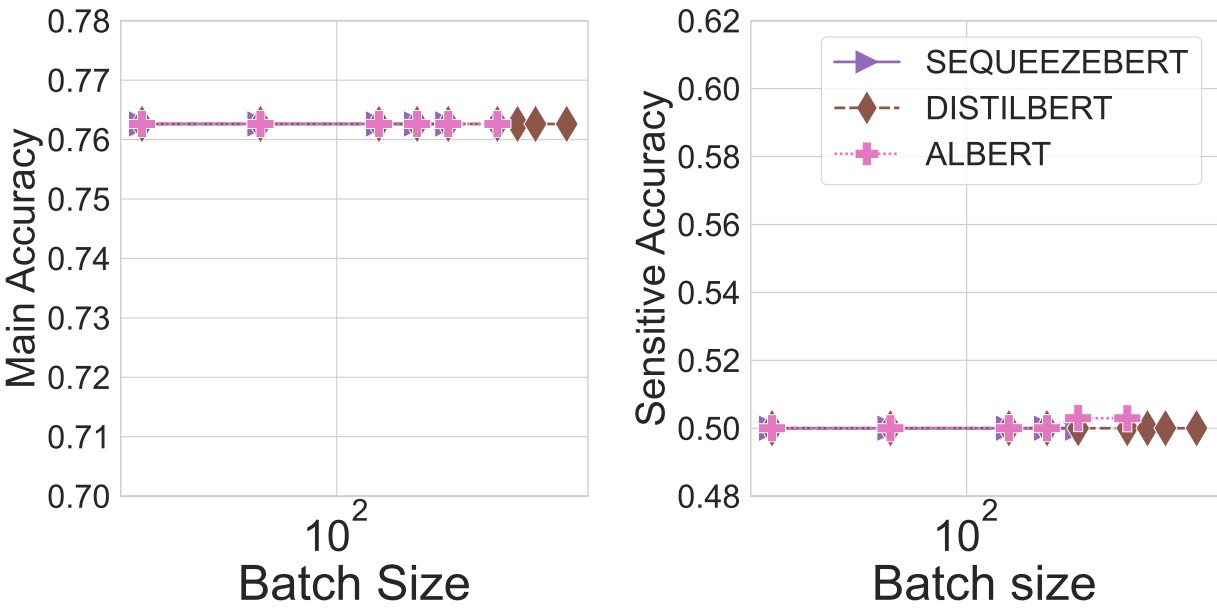

Fig. 13: Ablation study on batch size using `DIS.` for `TRUST-A`. Results on `TRUST-G` are given for $\lambda = 1$.

### B.3.2 Study on `PT` models

Fig. 14 gathers the results on the ablation study on `PT` for `TRUST`. We observe that for most of the considered `PT`, disentangling the representations improves the main task accuracy performance. We observe that overall `CLINIC` (with $\mathcal{S}_1$) achieves the best performances and can reach perfectly disentangled representations.

**Takeaways.** When changing the pretrained model, we observe a consistent behavior of `CLINIC` which outperforms the considered baselines. The observe improvement when disentangling the representation is

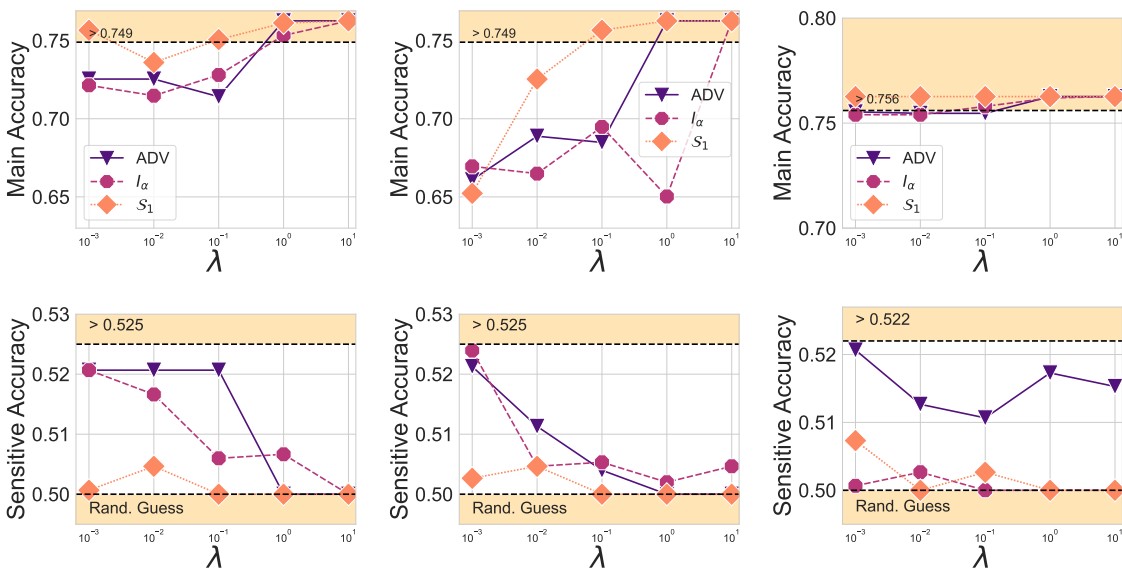

Fig. 14: Ablation study on pretrained models on `TRUST-A`. Figures from left to right corresponds to the performance of `ALB.`, `DIS.` and `SQU.` on `TRUST-G`.

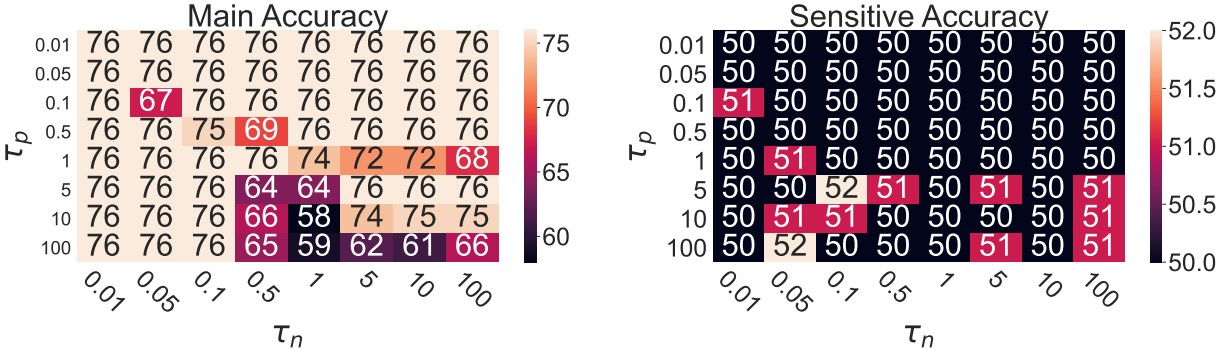

Fig. 15: Ablation study on $(\tau_n, \tau_p)$ for `TRUST-A`. Results are reported using `BERT` on `TRUST-G` for the main task $\uparrow$ (left) and the sensitive task $\downarrow$ (right) for $\lambda = 0.1$ and $B = 256$.

consistent across the different models which further validate the spurious correlation interpretation. This phenomenon is likely to be dataset and not model specific.

### B.3.3   Study on temperature

We report on Fig. 15, the results of the ablation study on temperature conducted on `TRUST-G`. When the sensitive attribute is easier to disentangle, we observe that the sensitive accuracy is less impacted by a change in the temperature as most of the models achieve perfect disentanglement. However, there is an impact on the main task accuracy. Overall we observe some $(\tau_n, \tau_p)$ area achieving lower results (bottom right of the matrix Fig. 15).

**Takeaways.** When working on `TRUST` that is easier to disentangle, we observe that the temperature tuning seems to have less impact on the sensitive task performance. However, it is needed to ensure a good sensitive/main task accuracy trade-off.

## C Training Details

We report in this section training details to reproduce the reported experiments. We give estimate run times on NVIDIA-V100 with 32Go of memory.

### C.1 Model architecture

We report in the section the considered model architectures.

**RNN-encoder.** For this model, we use a bidirectionnal GRU Chung et al. (2014) with 2 layers (the hidden dimension is set to 128) with LeakyReLU Xu et al. (2015). The encoder is followed by multi-layer perceptron of dimension 256.

**Pretrained-encoder.** For this model the encoder is followed by a single projection layer.

**Attacker network.** It is composed of 3 hidden layers with input sizes of $H_{dim}$, where $H_{dim}$ refers to the dimension of the attacked network.

### C.2 Implementation details

All the models have been trained using the `ADAMW` optimizer Loshchilov & Hutter (2017) which is an improvement of the `ADAM` Kingma & Ba (2014) optimizer with warmup set to 1000 Vaswani et al. (2017). The dropout rate Srivastava et al. (2014) has been set to 0.2. The learning rate of the optimizer has been set to 0.001 for the `RNN`-based encoders and to 0.00001 for the `BERT` models. The `PT` are trained for 15k iterations and the randomly initialized `RNN` are trained for 30k iterations.

**Used libraries.** In this work we relied on the following libraries and would like to thank their authors for open-sourcing them:

- Code from Barrett et al. (2019); Elazar & Goldberg (2018) to process `DIAL`. The baseline ADV has been re-implemented in our framework. The associated Github can be found `https://github.com/yanaiela/demog-text-removal`.

- Code from Coavoux et al. (2018) to process `DI`. In their work they also used the ADV baseline has been re-implemented in our framework. The associated GitHub can be found here: `https://github.com/mcoavoux/pnet`.

- Code from Colombo et al. (2021) has been privately shared by the authors.

- All the models have been implemented in Pytorch Paszke et al. (2017).

- The tokenizers, the `PT` have been taken from transformers Wolf et al..

### C.3 Computing costs

In this section, we report the (estimated) computational cost of reproducing our experiments. Note that we relieved on NVIDIA-V100 with 32 GO of memory. Each pretrained model takes (approximately) 3 hours to train and whereas the `RNN` encoder requires 5 hours to be trained. All models have been trained with batch size of 256. An adversary requires around 15 mins to be trained. **Controlling experiment.** For this experiments we trained $5 \times 5 \times 4 = 100$ `RNN` encoders and the same number of pretrained models. For each model we evaluate 6 checkpoints which makes 1200 probing classifiers. The checkpoint with lowest $\mathcal{R}$ (see Eq. 1) is selected. This experiment requires $500 + 300 + 1200/4 = 1100h$ of GPUs.

**Ablation study.** For the batch size experiment, we trained $8 \times 3 \times 2 = 48$ models with 240 probing classifiers for a total cost of GPU 204 hours. For the pretrained models ablation study, we trained $5 \times 3 \times 2 = 30$ models with 150 probing classifiers for a total cost of 128 hours. For the temperature we trained $8 \times 8 \times 2 = 128$ models with 640 probing classifiers for a total of 544 GPU hours.

**Overall summary.** The approximated total cost is around 2k GPU hours.

