# OpenReview forum: "A Novel Information Theoretic Objective to Disentangle Representations for Fair Textual Classification"
_TMLR — Rejected by TMLR_

### Review · Reviewer_AoBP · 2022-12-31

**Summary Of Contributions:**

The paper proposes a new approach for learning representations that are useful for classification while carrying little information about a sensitive attribute. To do that, the paper proposes to minimize the *conditional mutual information* between the representation and the sensitive attribute conditioned on the classification label, as opposed to prior work which minimizes the *mutual information* between the representation and the sensitive attribute. Experiments on several datasets show that the proposed approach achieves a good trade-off between the main task accuracy and the sensitive task accuracy.

**Audience:**

Yes

**Broader Impact Concerns:**

I do not have concerns about this.

**Claims And Evidence:**

No

**Requested Changes:**


Main concerns:

* Table 1: The presentation of this table is confusing. For "Y" score, a higher value is better. However, the bold numbers are not denoted according to this rule. For example, in DIAL-M dataset, CE achieves the best Y score, but S_1 and S_2 are marked in bold.

* In "Comparing the strategies of CLINIC." paragraph, it says "S1 and S2 are the best and always improve performance upon S0." Again, in DIAL-M dataset, S_0 has a better Y score than S_1 and S_2. The definition of "best" is not specified anywhere.

* Section 5.4: The complexity of the proposed approach scales quadratically to the batch size, but the comparison in this table is on one fixed batch size. It should comment on how batch size influences the comparison here.

* Table 2: what is the unit of the numbers in "1 epoch" column?

* The idea of penalizing conditional mutual information is nice. However, the final objective function is not doing this exactly. It should mention explicitly what is the gap between (the upper bound of) the objective (LHS of Eq. 7) and the conditional mutual information. In other words, it should explain Remark 2 in more detail.

Minor issues:

* Eq 1: \phi is already used in the first paragraph of Section 2. It is better to use different notations to reduce confusion.

* Section 3.2.2: it says "\bar{y} a uniformly sampled label in \mathcal{Y} \ \{y\}". However, according to the strategy description, \bar{y} is the label of the samples that satisfy the P_{S} and N_{S} definitions. It may be more accurate to denote \bar{y} = \mathcal{Y} \ \{y\}, and say y_{j_n} \in \bar{y} etc. in the strategy description (if I understand the algorithm correctly).

* Remark 1: what is \bar{S}?


**Strengths And Weaknesses:**


Strengths:

* The idea of penalizing conditional mutual information is interesting.

Weaknesses:

* The presentation is not clear in many places. See details below.

---

### Review · Reviewer_SVhx · 2023-01-01

**Summary Of Contributions:**

This paper aims at improving the disentanglement regularizer for fair classification, by replacing the widely used mutual information (MI) regularizer I(Z;S) with a conditional MI (CMI) term I(Z;S|Y), where Y, S, Z are classification labels, a sensitive attribute, and learned representations for simultaneously predicting Y and being disentangled with S, respectively. This paper is well organized, and easy to follow. In this paper, an algorithm to estimate the CMI term by contrastive learning is proposed, which seems novel and useful. Experiments to show the superiority of the proposed CMI regularizer are extensive. However, some claims about the existing regularizer and the proposed CMI term might be problematic in my analysis (see the following discussions) and are not convincing.



**Audience:**

Yes

**Claims And Evidence:**

No

**Requested Changes:**

see above.

**Strengths And Weaknesses:**

1） I think the first mistake occurs in Proposition 1 (Page 3), in which the Demographic Parity criterion is defined by P(\hat{Y}=y|S=0)= P(\hat{Y}=y’|S=1) for all y, y’. As far as I know, the Demographic Parity criterion should be defined by P(\hat{Y}=y|S=0)= P(\hat{Y}=y|S=1) for all y. Otherwise, the definition in this paper requires P(\hat{Y}) to be a uniform distribution, which is impossible for imbalanced classification. In addition, the conclusion of proof of Proposition 1 (Appendix A.1) should be P(\hat{Y}=y|S=0)= P(\hat{Y}=y|S=1).

2）The most problematic claim is the analysis about the minimization of I(Z;S) (Sec. 3.1.2, Page 4), which is the motivation of this paper. The authors claimed minimizing I(Z;S;Y) ‘could be problematic since I(Z;S;Y) tends to decrease the MI between Z and Y. This is a hasty conclusion. For fair classification, the ideal representations Z should eliminate some information related to S, to ensure the independence of Z and S (condition of Proposition 1), even though such information might be useful for predicting Y. The key question here is: is I(Z;S;Y) not related with S? The answer is obviously no, and hence the minimization of I(Z;S;Y) is necessary for fair classification. Theoretically, if I(Z;S;Y)>0, then I(Z;S)=I(Z;S|Y)+I(Z;S;Y)>= I(Z;S;Y)>0, which means Z will not be independent of S, and hence ensuring demographic parity via Proposition 1 is impossible. Therefore, the motivation of this paper is not convincing in my opinion. I will recommend the authors to carefully discuss the motivation and address its importance by theoretical analyses and experiments, rather than just passing it slightly.

3）My concerns about the motivation do not mean that the proposed method is also problematic. In fact, the proposed CMI is a weaker regularizer for fairness, but might have some unknown merits. To show the rationality of such regularizer, the authors should give a close look at it (and the induced algorithm) and provide a self-consistent story.

4）Some concepts mentioned in the paper lacks proper introductions and discussions, including demographic parity (Sec. 2.3, Page 3), Veyne diagrams (Sec. 3.1.1, Page 4), I(Z;S|Y) and I(Z;S;Y) (Sec. 3.1.2, Page 4), strategies S_1 and S_2 (Sec. 3.2.2, Page 5), etc. These concepts might be confusing to readers without background knowledge, and short introductions and discussions about their meanings are helpful to improve this paper’s readability.

5）The marks in Tab. 1 (Page 7) are misleading. How do you find the best model according to two different metrics, especially when it cannot achieve the best in the two metrics meanwhile? Besides, in TRUST-G dataset with RNN backbone, the baseline I_{alpha} is the best model according to Tab. 1, but why is it not bolded?

6）The setting of weight factor \lambda of the baseline method S_0 in Tab. 1 is probably not fair. As the penalization of I(Z;S) (medthod S_0) is stronger than I(Z;S|Y) (the proposed method S_1 and S_2), and hence its optimal weight factor is probably lower. This paper compares it with the proposed methods under the same weight factors, which is unreasonable.

Overall recommendation: reject / reject & resubmit

---

### Review · Reviewer_tH7u · 2023-01-07

**Summary Of Contributions:**

The paper proposes a new objective to learn the fair classifier. Specifically, the classic minimax objective is $I(Z,Y)-\lambda I(Z,S)$ where $Z$ is the representation $Y$ is the class distribution and $S$ is the sensitive attribute. The proposed objective is $I(Z,Y)-\lambda I(Z,S|Y)$.

**Audience:**

Yes

**Broader Impact Concerns:**

The paper focuses on improving the fairness of machine learning methods. One potential impact of such a paper could be to improve the fairness and bias-mitigation capabilities of machine learning algorithms. This could lead to more accurate and unbiased predictions and decisions made by these algorithms, which could have a wide range of applications in areas such as credit scoring, hiring, and criminal justice.



**Claims And Evidence:**

Yes

**Requested Changes:**

1. Explain why the proposed objective is better scientifically, e.g. has a better global optimal.
2. Find the best $\lambda$s for both objectives, and compare the results using the individual's best $\lambda$ rather than using the same $\lambda$.
3. A detailed derivation of section 3.2.1 to make the paper self-contained.

**Strengths And Weaknesses:**

Strength: the paper is easy to follow.

Weakness:
1. It's not clear to me if the proposed objective is well-motivated. The paper claims that minimizing  $I(Z,S)=I(Z,S|Y)+I(Z,S,Y)$ will also minimize $I(Z,S,Y)$ which contains the mutual information $I(Y,Z)$. However, since $S->Z->Y$ forms a Markov chain,  $I(S,Y|Z)=0$ and $I(Z,S,Y)=I(S,Y)-I(S,Y|Z)=I(S,Y)$, which I guess is the quantity we want to minimize.
2. In practice, since the mutual information is intractable and requires approximation, minimising $I(Z,S)$  may reduce $I(Y,Z)$ in practice.
In this case, you can always make the $\lambda$ smaller to offset the effect, and both objectives will have the same optimal point.

---

### Decision · Action_Editors · 2023-02-20

**Recommendation:** Reject

**Comment:**

This paper presents an interesting idea: feature disentanglement using the information gain for fair (textual) classifiers. All the reviewers are concerned about the theoretical justifications and some unfair experimental settings. Unfortunately, the authors didn't provide any feedback. After discussion with the reviewers, they all recommended reject.

**Audience:**

This paper is 0f general interest to TMLR readers.

**Claims And Evidence:**

Some of the key claims are misleading and unclear.